# Probing the effect of clustering on EphA2 receptor signaling efficiency by subcellular control of ligand-receptor mobility

Zhongwen Chen[1,2†], Dongmyung Oh[3,4†], Kabir Hassan Biswas[5*], Ronen Zaidel-Bar[6*], Jay T Groves[2*]

[1]Multiscale Research Institute of Complex Systems, Fudan University, Shanghai, China; [2]Department of Chemistry, University of California, Berkeley, Berkeley, United States; [3]Department of Biochemistry and Molecular Biology, The University of Texas Medical Branch at Galveston, Galveston, United States; [4]Mechanobiology Institute, National University of Singapore, Singapore, Singapore; [5]College of Health and Life Sciences, Hamad Bin Khalifa University, Doha, Qatar; [6]Department of Cell and Developmental Biology, Sackler Faculty of Medicine, Tel Aviv University, Tel Aviv, Israel

**Abstract** Clustering of ligand:receptor complexes on the cell membrane is widely presumed to have functional consequences for subsequent signal transduction. However, it is experimentally challenging to selectively manipulate receptor clustering without altering other biochemical aspects of the cellular system. Here, we develop a microfabrication strategy to produce substrates displaying mobile and immobile ligands that are separated by roughly 1 μm, and thus experience an identical cytoplasmic signaling state, enabling precision comparison of downstream signaling reactions. Applying this approach to characterize the ephrinA1:EphA2 signaling system reveals that EphA2 clustering enhances both receptor phosphorylation and downstream signaling activity. Single-molecule imaging clearly resolves increased molecular binding dwell times at EphA2 clusters for both Grb2:SOS and NCK:N-WASP signaling modules. This type of intracellular comparison enables a substantially higher degree of quantitative analysis than is possible when comparisons must be made between different cells and essentially eliminates the effects of cellular response to ligand manipulation.

*For correspondence:
kbiswas@hbku.edu.qa (KHB);
zaidelbar@tauex.tau.ac.il (RZ-B);
JTGroves@lbl.gov (JTG)

[†]These authors contributed equally to this work

Competing interests: The authors declare that no competing interests exist.

## Introduction

The cell membrane surface is studded with a broad array of receptor proteins that interact with numerous ligands, which can be soluble, membrane-bound on an apposed cell surface, or associated with the extracellular matrix (*Casaletto and McClatchey, 2012*; *Downward, 2001*; *Groves and Kuriyan, 2010*). Ligand binding is followed by activation of elaborate signal transduction pathways in the cell, which ultimately mediate cellular decision-making. In the case of receptor tyrosine kinases (RTKs), initial receptor activation after ligand binding involves phosphorylation of specific tyrosine residues on the receptors, followed by recruitment of adaptor proteins which mediate activation of additional signaling molecules. The membrane receptors, adaptors, and signaling molecules form complexes to induce immediate local responses, such as actin polymerization, or transduce signals into nucleus, such as through mitogen-activated protein kinase (MAPK) signaling cascade. Assembly of cell surface receptors into clusters or organized arrays is a common feature of cell membranes (*Dustin and Groves, 2012*; *Garcia-Parajo et al., 2014*; *Janes et al., 2012*; *Lee et al., 2002*;

*Mossman et al., 2005*; *Salaita et al., 2010*), and has long been implicated as an important factor for modulating signaling activity (*Bray et al., 1998*; *Cebecauer et al., 2010*; *Jadwin et al., 2016*; *Oh et al., 2012*).

More recently, receptor clustering in some cases has also been found to involve downstream signaling molecules that appear to undergo protein condensation phase transitions (*Banjade and Rosen, 2014*; *Case et al., 2019*; *Huang et al., 2019*; *Huang et al., 2017b*; *Huang et al., 2016*; *Li et al., 2012*; *Su et al., 2016*). For example, in the T cell receptor signaling system, linker for activation of T cells (LAT) phosphorylation results in the recruitment of the adaptor protein Grb2 through binding to multiple phosphorylated tyrosine sites on LAT. Grb2 additionally possesses two SH3 domains that bind the guanine nucleotide exchange factor, Son of Sevenless (SOS), with the latter also serving as a crosslinking molecule to bridge multiple Grb2 and LAT molecules. The resulting LAT:Grb2:SOS assembly can form a two-dimensional bond percolation network, or protein condensate, on the cell membrane (*Huang et al., 2016*; *Su et al., 2016*). Recent experimental investigations have revealed that the physical form of this network prolongs the dwell time of SOS molecules on membrane, which ultimately establishes a kinetic proofreading mechanism to modulate the activation of SOS and its downstream effector Ras GTPase (*Huang et al., 2019*). In a similar fashion, the nephrin receptor crosslinks the adaptor proteins NCK and neural Wiskott-Aldrich syndrome protein (N-WASP) to form a protein condensate (*Li et al., 2012*), which is crucial for N-WASP activation by a similar mechanism as observed with SOS (*Case et al., 2019*). These discoveries advance a mechanistic understanding of how the spatial assembly of signaling molecules can modulate the chemical outcomes of the system. Furthermore, it is becoming increasingly clear that such signaling assemblies likely play an instrumental role in defining the organization of the cell membrane (*Chung et al., 2021*; *Freeman et al., 2018*; *Kalappurakkal et al., 2019*).

Although receptor assembly and phase transition processes are evident in cellular systems, it remains difficult to precisely define their functional consequences on signaling itself in living cells. A major reason for this is that chemical perturbation of assemblies, such as those achieved with pharmacological agents or mutations (*Bugaj et al., 2013*; *Bugaj et al., 2015*; *Davis et al., 1994*; *Schaupp et al., 2014*; *Seiradake et al., 2013*; *Su et al., 2016*; *Wu et al., 2015*), are likely to produce side effects on the cell other than modulating molecular assembly itself. For example, phosphorylation at multiple tyrosine sites on the LAT protein mediates multivalent binding of Grb2 and SOS. Altering the number of these tyrosine sites by point mutations changes LAT clustering levels and subsequent signaling responses (*Houtman et al., 2006*; *Huang et al., 2017a*; *Su et al., 2016*). In such experiments, however, the contribution of clustering itself can not generally be separated from effects of the total copy number of involved signaling molecules (which is also changing) as well as other overall changes in cellular responses stemming from altered signaling activity. In fact, this problem is universal in biophysical studies of protein organization in living cells: any perturbation that alters biological function will tend to produce a cascade of cellular effects, which can be very difficult to disentangle. An additional factor that compounds the complexity of signaling outcomes in studies involving a population of cells is the inherent cell-to-cell heterogeneity. These mandate studying individual cells in order to draw robust conclusions on quantitative aspects of signaling.

To address these issues, we seek to modulate receptor spatial organization by controlling ligand mobility, instead of perturbing intracellular components such as by creating genetic mutations or pharmacological treatments. In this respect, we have previously developed a microfabrication strategy to produce micropatterned hybrid substrates displaying ligands on both immobile, surface adsorbed polymers and mobile supported lipid membranes to simultaneously reconstitute cell-matrix and cell-cell interactions in individual cells (*Chen et al., 2018*). The multicomponent, micropatterned supported membrane substrates also allow for spatially segregated functionalization of different ligands to dissect crosstalk between different RTKs (*Biswas et al., 2018b*; *Biswas et al., 2018a*), in a way that is not feasible with non-patterned supported membrane substrates (*Biswas, 2020*; *Biswas and Groves, 2019*; *Biswas et al., 2015*; *Biswas et al., 2016*; *Vafaei et al., 2017*; *Yu et al., 2015*). Here, we extend this method to display a ligand of interest in both mobile and immobile configurations, spatially juxtaposed on length scales small enough to enable a side-by-side comparison within an individual living cell. The immobile ligands are displayed on a functionalized poly L-Lysine-poly ethylene glycol (PLL-(*g*)-PEG) scaffold; while the mobile ligands are displayed on supported lipid membrane corrals which allows cluster formation. We employ an in situ

UV-ozone lithographic process to selectively remove regions of the PLL-(*g*)-PEG substrate and subsequently replace these with supported lipid membranes.

Key to this strategy is that the clustered and non-clustered receptors are separated by a few microns and thus experience an identical cytoplasmic signaling state, enabling precision comparison of downstream signaling reactions. In this work, we utilize the micropatterned hybrid substrates to study the ephrinA1:EphA2 signaling system in living cells. EphA2 is a member of the Eph RTK family which is often overexpressed in aggressive breast cancers (*Fox and Kandpal, 2004*; *Macrae et al., 2005*). Clustering is thought to be integral to the ligand-mediated activation of EphA2 and downstream signaling (*Davis et al., 1994*; *Salaita et al., 2010*). When ephrinA1 is displayed on a mobile supported membrane surface, clustering upon interaction with EphA2 expressed on the surface of a living cell is readily observed. Here, we reconstituted ephrinA1:EphA2 interactions in both mobile and immobile configurations in individual cells to study the effects of receptor clustering on signaling transductions. Using a series of live-cell imaging and single-molecule tracking experiments, we monitored local signaling events in a spatially resolved manner. The results show that clustering significantly increases EphA2 signaling efficiency by inducing greater receptor phosphorylation and subsequent Erk activation and actin polymerization. Both Grb2:SOS and NCK:N-WASP assemblies were observed in EphA2 clusters. Importantly, the receptor clustering consistently increases molecular binding dwell times, which results in a modulation of their functions. Thus in accordance with other in vitro reconstitution studies (*Case et al., 2019*; *Huang et al., 2019*), our results provide further evidence to support the kinetic proofreading mechanism of molecule activation in biomolecular condensates.

## Results

### EphrinA1 displayed on micropatterned membrane corrals are clustered by cells

With the ultimate goal of elucidating the role of receptor clustering in cellular signaling, we started by microfabricating substrates containing micron-scale supported membrane corrals displaying mobile ephrinA1 to reconstitute ephrinA1:EphA2 juxtacrine complexes on synthetic substrates (*Figure 1A*; *Chen et al., 2018*). For this, biotinylated poly-l-(lysine)-grafted-polyethylene (glycol) polymer (PLL-(*g*)-PEG-Biotin) was first coated on a glass coverslip, followed by selective deep UV lithography with a photomask (*Figure 1—figure supplement 1A*). The deep UV-exposed PLL-(*g*)-PEG-Biotin was efficiently degraded and washed away, as ascertained by atomic force microscopy measurement after this step (*Figure 1B*). Supported membranes were then assembled on the cleared glass area by vesicle fusion method, and subsequently functionalized with Alexa680-labeled ephrinA1 via Ni-NTA:poly-histidine interaction (*Nye and Groves, 2008*). In addition, biotinylated cyclic RGD peptide was functionalized to the PLL-(*g*)-PEG-biotin through Dylight405-labeled NeutrAvidin conjugation. RGD can engage integrins to allow cell spreading, during which cells can interact with multiple ephrinA1-functionalized supported membranes (*Chen et al., 2018*). Fluorescent images confirmed successful functionalization of the supported membranes with ephrinA1, and their spatial segregation from RGD areas (*Figure 1B*). Importantly, ephrinA1 was mobile on the supported membranes, as shown by fluorescence recovery after photobleaching (*Figure 1—figure supplement 1B*).

We then monitored the interaction of live cells with ephrinA1 on supported membrane corrals. MDA-MB-231 breast cancer cell line was used as a model cell here, since it expresses EphA2 at a high level and has been characterized previously on ephrinA1 functionalized supported membranes (*Chen et al., 2018*; *Greene et al., 2014*; *Lohmüller et al., 2013*; *Salaita et al., 2010*; *Xu et al., 2011*). EphrinA1 surface density on supported membranes was calibrated with a quantitative fluorescence measurement (*Galush et al., 2008*; *Figure 1—figure supplement 1C*), and was controlled to be around 100 molecules/$\mu m^2$ for these experiments, which is comparable with EphA2 expression levels in cell lines (*Salaita et al., 2010*; *Xu et al., 2011*). Cells seeded on the micropatterned substrate spread presumably through engagement of integrins on the cell membrane with RGD ligands on the substrate. This dynamic spreading enabled cell membrane protrusions to physically interact with multiple ephrinA1 membrane corrals on the substrate, resulting in the clustering of diffusive ephrinA1 ligands through binding of cellular EphA2 receptors (*Figure 1C* and *Video 1*).

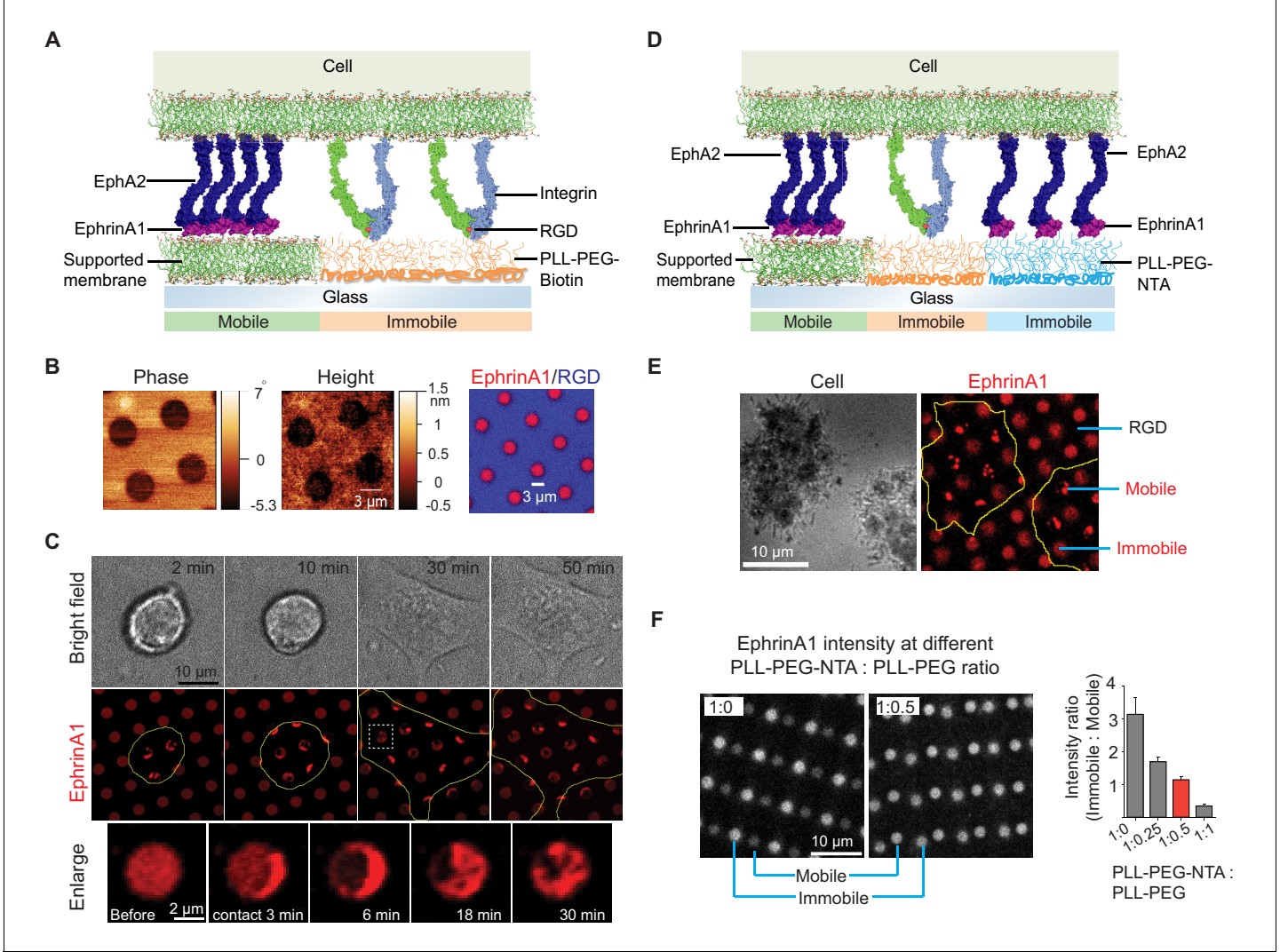

**Figure 1.** Presentation of mobile and immobile ephrinA1 ligands to a single cell. (**A**) Schematic illustration of the two-component substrate. (**B**) AFM and fluorescence images of the substrate. (**C**) Representative bright-field and fluorescence images of a cell spreading on the substrate with indicated spreading time. Yellow outlines indicate the cell boundary. Bottom panel represents an enlarged ephrinA1 membrane corral before or after cell contact. (**D**) Schematic illustration of the three-component substrate. (**E**) Representative reflective interference contrast and fluorescence images of cells spreading on the substrate. (**F**) Fluorescence images (in gray scale) and quantification of ephrinA1 intensity in mobile or immobile corrals by titrating PLL-(*g*)-PEG-NTA with PLL-(*g*)-PEG. Data are presented as mean± SD. AFM, atomic force microscopy.

The online version of this article includes the following figure supplement(s) for figure 1:

**Figure supplement 1.** Micropatterning of 2-component substrate.

**Figure supplement 2.** Micropatterning of 3-component substrate.

The clustering of ephrinA1 is relatively fast. Almost all the ligands inside each corral were clustered within 6 min of physical contact by the cell membrane, leading to the formation of typically a micro-cluster in each of the corrals (*Figure 1C*). The cluster is also dynamic, moving from one side to another and occasionally splitting into smaller clusters, resulting in a great degree of variability as shown by the large standard deviations of clustered ephrinA1 intensity in each membrane corral (*Figure 1—figure supplement 1D*). Because of the spatial confinement and forces applied by the cell (*DeMond et al., 2008*; *Mossman et al., 2005*; *Salaita et al., 2010*), clusters tend to be trapped in the periphery of the corrals. This system allows for spatiotemporally resolved observation of ephrinA1 interaction with EphA2, as well as its downstream signaling events. The supported membranes in the micropatterned substrates can also be functionalized with other ligands, for example,

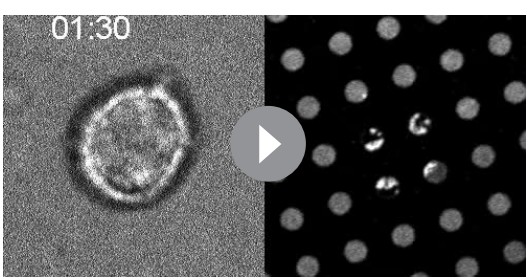

**Video 1.** Bright-field and fluorescence images of an MDA-MB-231 cell spreading on ephrinA1 micropatterned substrate. Left: bright field. Right: ephrinA1.
https://elifesciences.org/articles/67379#video1

E-cadherin to form cadherin junctions (*Figure 1—figure supplement 1E*), demonstrating versatility of this technology to study different receptors.

## Spatially segregated display of mobile and immobile ephrinA1 for single cells

Next, the substrates were extended to contain both mobile and immobile ephrinA1 to study the effects of ligand clustering on receptor signaling (*Figure 1D*). For this, PLL-(*g*)-PEG-biotin coated glass substrate was selectively UV-etched and was coated with another polymer PLL-(*g*)-PEG-NTA. This hybrid substrate then underwent a second UV etch to form supported membranes, which led to a three-component substrate (*Figure 1—figure supplement 2A*). The sequential UV etch processes can be overlaid randomly for regular circular arrays, because it is able to generate sufficiently large areas with clearly separated mobile and immobile regions in a centimeter-size pattern. However, the glass coverslip and the photomask can also be aligned under the microscope before each UV exposure for the accurate layout of multiple components as needed, at a cost of extra time and alignment instrument (*Figure 1—figure supplement 2B*).

EphrinA1 was functionalized on both PLL-(*g*)-PEG-NTA polymers and supported membranes through the same Ni-NTA:poly-histidine interactions, while RGD was functionalized on PLL-(*g*)-PEG-Biotin to allow cells to spread. *Figure 1E* shows an example of cells spreading on the substrate with alternating mobile ephrinA1 membrane corrals and immobile ephrinA1 polymers. Clearly, only mobile ephrinA1 became clustered after contact with cells. We found that using undiluted PLL-(*g*)-PEG-NTA gave a higher ephrinA1 density in immobile regions compared to mobile regions. However, diluting the polymer with non-reactive PLL-(*g*)-PEG enabled us to tune the ephrinA1 density to be similar in both mobile and immobile regions, in the range of 50–100 molecules/$\mu m^2$ (*Figure 1F*).

## Mobile ephrinA1 increases EphA2 phosphorylation through clustering

Display of mobile and immobile ephrinA1 on micron-scale corrals (membrane and polymer, respectively) allowed for comparison of clustered and non-clustered EphA2 receptor signaling in a spatially resolved manner in individual cells. Immunostaining of cells using an antibody against the intracellular domain of EphA2 verified engagement of the receptors to both mobile and immobile ephrinA1 (*Figure 2A*). Although EphA2 cluster formation is dependent on the ligand mobility, the total amount of EphA2 receptors recruited to mobile or immobile ephrinA1 were very similar, with a slight decrease in mobile ephrinA1 regions possibly due to endocytosis (*Greene et al., 2014*; *Sugiyama et al., 2013*; *Figure 2B*, *Figure 2—source data 1*), suggesting a conservation of binding between receptors and ligands at the cell:substrate interface. EphA2 is known to undergo a ligand binding-induced autophosphorylation at tyrosine 588, and this phosphorylation site is key to the recruitment of downstream signaling molecules (*Parri et al., 2005*). Immunostaining of cells using an anti-pY588-EphA2 specific antibody showed that EphA2 phosphorylation increased by an average of 60% in response to mobile ephrinA1 stimulation compared to immobile stimulation (*Figure 2C and D*, *Figure 2—source data 1*). Consistent with ephrinA1 cluster variability, the fluorescence intensities of EphA2 (*Figure 2—figure supplement 1A*) and pY588-EphA2 (*Figure 2—figure supplement 1B*) also showed larger fluctuations in mobile regions compared with immobile ones. These results suggest that clustering of ephrinA1:EphA2 complexes enabled by the mobile ligands resulted in a change in their physicochemical properties: the clustering enhances EphA2 autophosphorylation, and in the meanwhile condenses its downstream molecules.

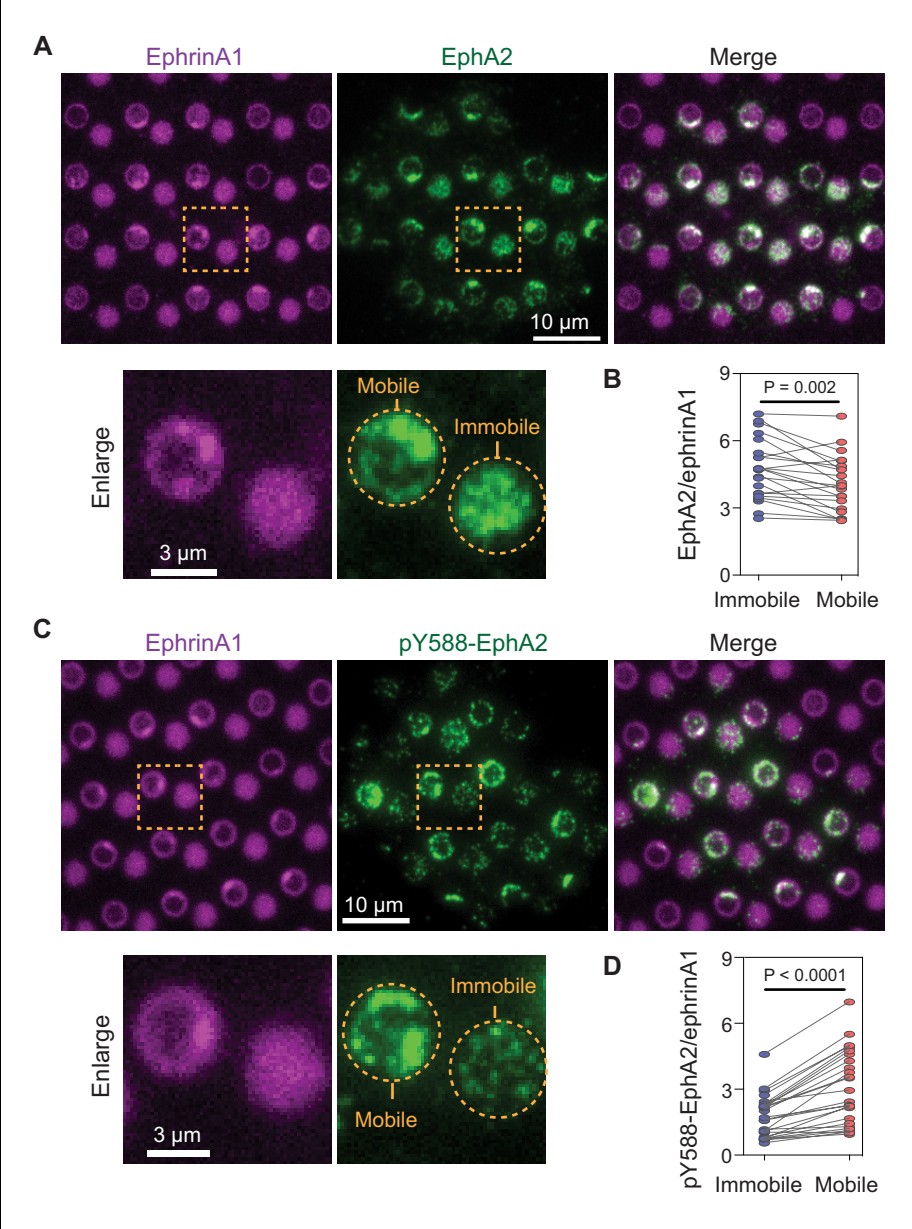

**Figure 2.** Mobile ephrinA1 stimulation increases EphA2 phosphorylation. (**A**) Representative immunofluorescence images of EphA2 in MDA-MB-231 cells fixed after 45 min spread on the substrate. The yellow square marked region is enlarged in bottom panel. (**B**) Quantification of EphA2/ephrinA1 intensity ratio in mobile and immobile ephrinA1 corrals. Each data point represents an averaged ratio from multiple corrals from a single cell. The two grouped data from the same cell are paired for comparison. Significance is analyzed by paired-group Student's t-test. N=22 cells. (**C**) Representative immunofluorescence images of pY588-EphA2. Condition same as above. (**D**) Quantification of pY588-EphA2/ephrinA1 intensity ratio in mobile and immobile ephrinA1 corrals. N=26 cells. The online version of this article includes the following source data and figure supplement(s) for figure 2:

**Source data 1.** Quantification of EphA2/eprinA1 or pY588EphA2/eprinA1 intensity ratio in mobile or immobile ephrinA1 regions.

**Figure supplement 1.** EphA2 clusters have large variability.

# EphA2 clustering enhances Grb2:SOS signaling transductions by increasing on-rate and molecular dwell time

Increased phosphorylation of EphA2 observed on mobile ephrinA1 membrane corrals is expected to lead to increased recruitment of intracellular effector proteins. Therefore, we monitored local signaling transmission events in clustered or non-clustered EphA2 receptors in single living cells. Grb2 is an important cytosolic adaptor protein that is known to be recruited to EphA2 receptors after ligand binding-induced phosphorylation (*Pratt and Kinch, 2002*). Grb2 further recruits SOS, which catalyzes Ras-GDP exchange to Ras-GTP, and activates the MAPK pathway (*Figure 3A*). For this, MDA-MB-231 cells transfected with Grb2-tdEos were seeded on the substrates to allow live imaging of Grb2 recruitment by total internal reflection fluorescence (TIRF) microscopy (*Video 2*). A significant difference was observed in the local recruitment of Grb2 to mobile or immobile ephrinA1:EphA2 complexes as cells spread (*Figure 3B and C*, *Figure 3—source data 1*). For the same number of ephrinA1 molecules, the mobile ligands increased Grb2 recruitment by about 80% compared to immobile ones both in terms of maximal (*Figure 3D*) and 30 min cumulative signals (*Figure 3E*), which is more prominent than the enhancement of receptor phosphorylation described above.

Grb2 and SOS assembly increases their molecular dwell time on the membrane, which forms a gating mechanism to control SOS activation in the reconstitution system (*Huang et al., 2019*; *Huang et al., 2016*). In our living cell system, the simultaneous presentation of clustered and non-clustered EphA2 in a single cell enables a precise comparison of Grb2 dwell time (or off-rate, $K_{off}=1/$ dwell time) by single-molecule imaging. Grb2-tdEos fluorescence emission spectrum can be photo-switched by UV excitation. A short UV (405 nm) exposure resulted in the illumination of a small amount of Grb2-tdEos in the red channel, which was then imaged at a rate of 20 frames per s (*Video 3*). The Grb2-tdEos particles were confirmed to be single molecules as shown by single-step photobleaching (*Figure 3—figure supplement 1A*), and the fluorescence intensities of individual particles also exhibited a unimodal distribution (*Figure 3—figure supplement 1B*). The apparent diffusion of Grb2-tdEos molecules was largely confined due to restricted EphA2 receptors either through clustering on mobile areas or by the fixed ligands on immobile areas (*Figure 3—figure supplement 1C*). The coordinates of every Grb2 molecule in a continuous movie were then assembled to generate a high-resolution localization image (also termed single-particle tracking Photo-Activation Localization Microscopy; sptPALM *Manley et al., 2008*), which clearly showed clustering of Grb2 in mobile ephrinA1 corrals while relatively uniformly distributed in immobile corrals (*Figure 3F*). By mask-separating these two regions, we found that Grb2 has a longer dwell time distribution on clustered ephrinA1:EphA2 complexes, in comparison to the non-clustered ones (*Figure 3G*). A membrane localized CAAX-tdEos control was used to measure the photobleach rate under the same experimental condition, which was significantly slower than the apparent Grb2 binding dynamics. The dwell time difference is consistent in all measured cells as shown by pairwise comparison of mobile or immobile regions in each individual cell. On

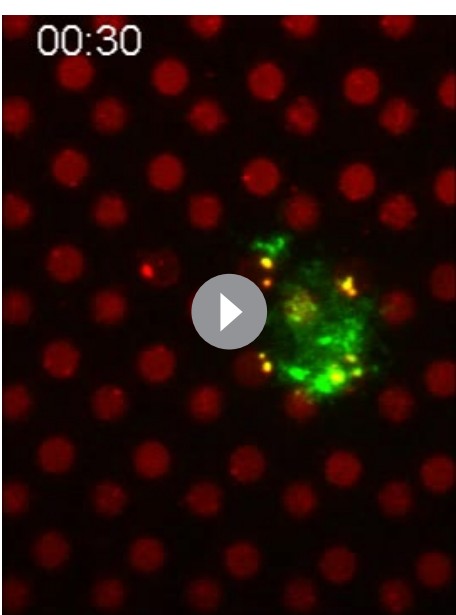

**Video 2.** A Grb2-tdEos transfected MDA-MB-231 cell spreading on ephrinA1 micropatterned substrate.
https://elifesciences.org/articles/67379#video2

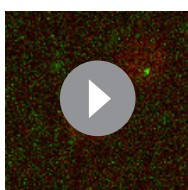

**Video 3.** Single-molecule imaging of Grb2-tdEos on mobile and immobile ephrinA1 substrate. Video played at real-time speed.
https://elifesciences.org/articles/67379#video3

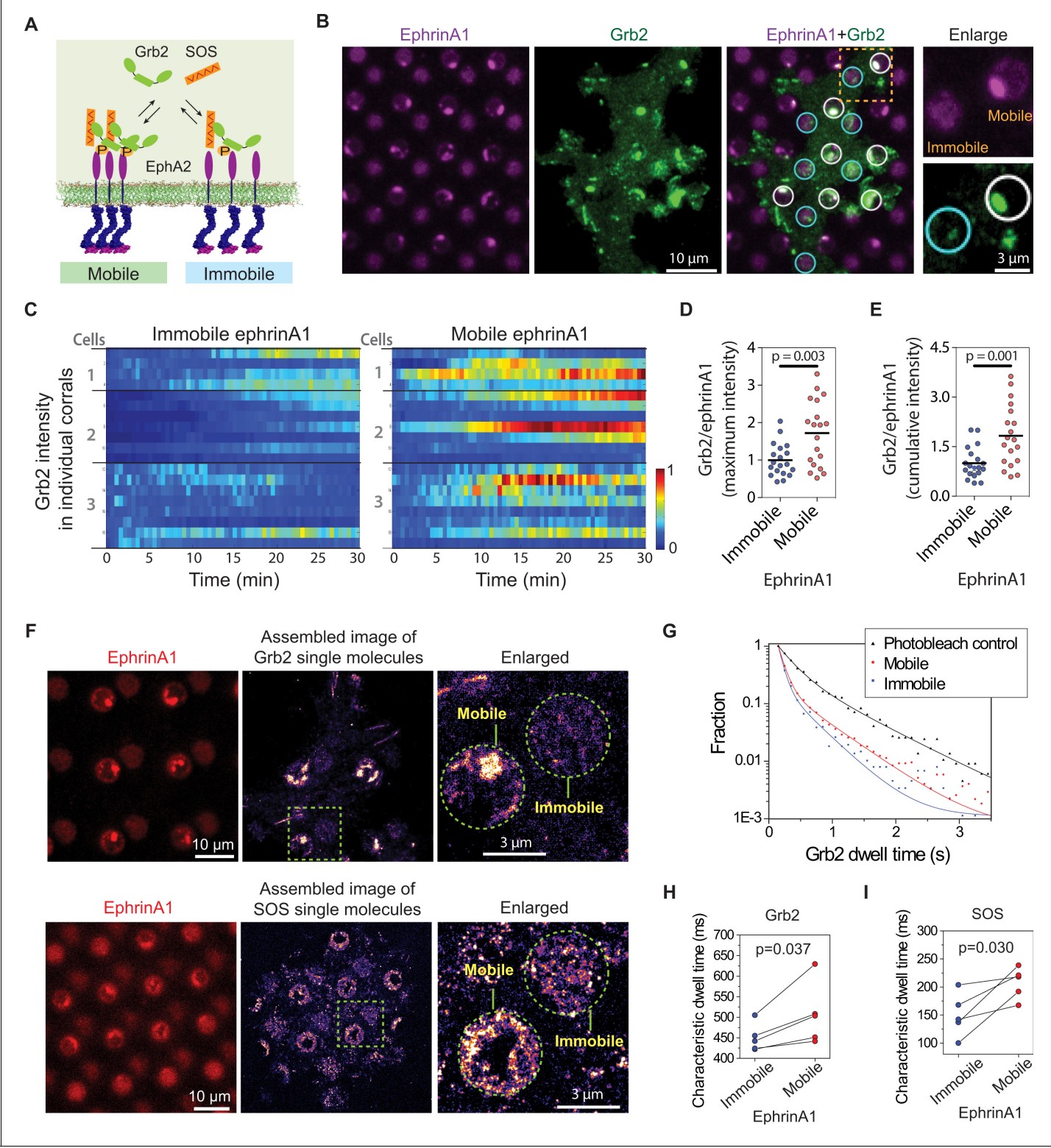

**Figure 3.** EphA2 clustering enhances Grb2:SOS signaling transductions by increasing on-rate and molecular dwell time. (**A**) Schematic illustration of Grb2 and SOS recruitment to mobile or immobile EphA2 receptors under the same cell. (**B**) Representative live-cell images of a Grb2-tdEos transfected cell spreading on the substrate after 45 min, with white circles indicating mobile ephrinA1 corrals and cyan circles indicating immobile ephrinA1. A yellow square marked region is enlarged to highlight the differential Grb2 recruitment to a mobile ephrinA1 corral and an immobile one. (**C**) Heat map of temporal Grb2-tdEos intensities in three cells. Each block represents the normalized intensity of Grb2 in an ephrinA1 corral at a given time, starting

*Figure 3 continued on next page*

*Figure 3 continued*

from cell contact to a total period of 30 min, at 30 s/frame acquisition speed. The intensity of Grb2 is normalized according to its highest intensity of all corrals through the whole time period for each cell, and is color-coded for visualization. Quantification of (**D**) maximum Grb2 intensity or (**E**) 30 min cumulative intensity in mobile or immobile ephrinA1 corrals, normalized with ephrinA1 ligand intensity in each corral. Significance is analyzed by Student's t-test. (**F**) Single-molecule imaging of Grb2-tdEos or SOS-tdEos. The coordinates of Grb2 or SOS single molecules when they first appear in a continuous movie are assembled to generate a localization image. (**G**) Distribution of Grb2-tdEos single-molecule dwell time. Membrane located CAAX-tdEos is applied as a photobleach control measured from another cell. Quantification of (**H**) Grb2-tdEos or (**I**) SOS-tdEos single-molecule dwell time. The dwell time distribution is fitted by a two-order exponential decay function and the slower time constant τ2 is used to represent characteristic dwell time for pairwise comparison in a group of cells. N=5 cells. Significance is analyzed by paired-group Student's t-test.

The online version of this article includes the following source data and figure supplement(s) for figure 3:

**Source data 1.** Time-lapse recording of Grb2-tdEos fluorescence intensity in each mobile or immobile ephrinA1 region.

**Figure supplement 1.** Single molecule imaging of Grb2-tdEos.

average it increased Grb2 dwell time from 451 ms in immobile areas to 508 ms in mobile areas (*Figure 3H*). Notably, the cell-to-cell variations of dwell time are in a similar level to the mobile-to-immobile differences; therefore, only a side-by-side comparison of each individual cell made it possible to detect the increase of Grb2 membrane dwell time induced by clustering. By counting molecule appearances in mobile or immobile regions, we found that clustering increases Grb2 association rate (Kon) by about 60%, a value commensurate with the increase of phosphorylated EphA2 (*Figure 3—figure supplement 1D*). As a result, the increase of 80% Grb2 total recruitment is a combination of increased receptor phosphorylation and elongated dwell time.

Similar to the observations with Grb2, EphA2 clustering also consistently increased SOS-tdEos single-molecule dwell time on the membrane. On average, it increased SOS dwell time from 150 ms in immobile areas to 207 ms in mobile areas (*Figure 3F and I*). Taken together, these data indicate that EphA2 clustering not only increases Grb2 and SOS recruitment by inducing higher receptor phosphorylation, but also increases their membrane dwell times, all of which contributing to the enhanced signaling transduction from ephrinA1:EphA2 complexes.

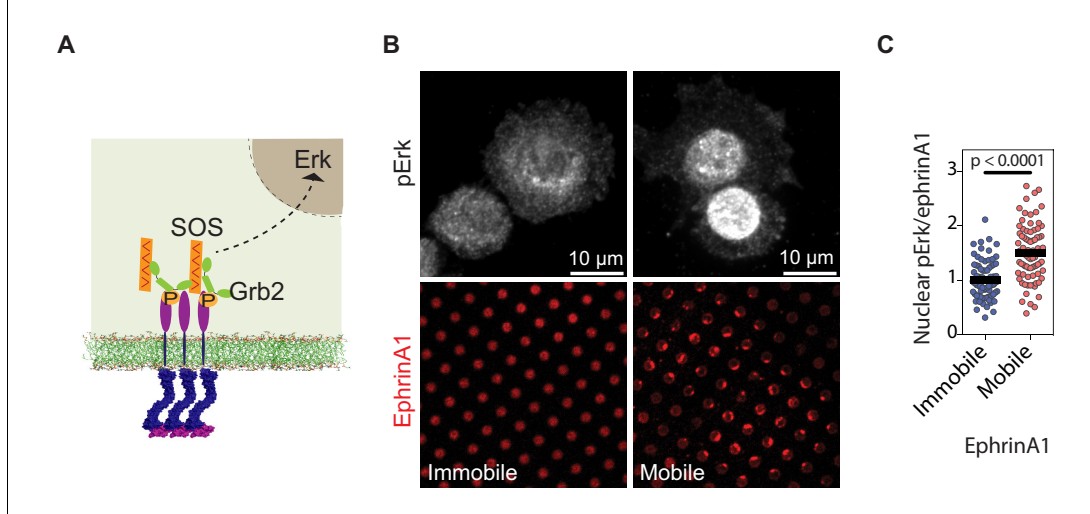

**Figure 4.** EphA2 clustering activates Erk. (**A**) Schematic illustration of the signaling pathway from EphA2 receptor to nuclear Erk activation. (**B**) Representative immunofluorescence images of pErk in PC-3 cells fixed after 45 min spread on the two substrates. Mobile ephrinA1 on supported membranes, or immobile ephrinA1 on PLL-PEG-NTA polymers, are respectively micropatterned on RGD background. (**C**) Quantification of nuclear pErk/EphrinA1 intensity ratio of cells fixed after 45 min spread on mobile or immobile ephrinA1 substrates. N=103 cells for mobile and 82 cells for immobile substrates. Significance is analyzed by Student's t-test.

The online version of this article includes the following source data for figure 4:

**Source data 1.** Quntification of nuclear pErk/ephrinA1 intensity ratio.

## EphA2 clustering enhances nuclear Erk activation

We further characterized the propagation of EphA2 membrane into nucleus through MAPK signaling pathway (*Figure 4A*). Since it is not possible to separate the phosphorylated nuclear Erk signal arising from mobile or immobile ephrinA1 in the same cell, we used separate substrates containing either mobile or immobile ephrinA1 at similar densities (~100 molecules/$\mu m^2$). Further, because MDA-MB-231 cell expresses a constitutively active Ras mutant, which is upstream of Erk; this experiment was performed using another cell line PC-3, which expresses a high level of EphA2 and is also sensitive to MAPK signaling (*Gregg and Fraizer, 2011*). Immunostaining of the PC-3 cells with a phospho-specific anti-Erk antibody revealed ~50% increase in the levels of activated nuclear Erk in cells seeded on mobile ephrinA1 substrates compared to immobile ones (*Figure 4B and C*, *Figure 4—source data 1*). This establishes that clustering of EphA2 upon binding bilayer-bound, mobile ephrinA1 results in an enhanced receptor signaling leading to higher adaptor and effector recruitment and subsequent Erk activation.

## EphA2 clustering enhances NCK:N-WASP induced actin polymerization and increases molecular dwell time

Actin polymerization is another important signaling module that has been studied extensively (*Banjade and Rosen, 2014*; *Case et al., 2019*; *Li et al., 2012*). It was found that in the reconstitution system, the receptor phosphorylation-dependent NCK:N-WASP assembly facilitates Arp2/3 complex activation and subsequent actin polymerization. Importantly, the N-WASP dwell time determines the strength of actin polymerization (*Case et al., 2019*). EphA2 has also been reported to modulate actin cytoskeleton (*Mohamed et al., 2012*). Here we sought to test if such molecular timing mechanism applies to EphA2 signaling in living cells (*Figure 5A*).

Similar to Grb2, NCK-mEOS3.2 and N-WASP-mEOS3.2 were clearly found to be enriched at ephrinA1:EphA2 clusters in comparison to immobile ephrinA1 regions (*Figure 5B*). Live imaging of F-tractin-EGFP showed local actin polymerization in each mobile ephrinA1 corral shortly after cluster formation; however, no enrichment of F-actin can be resolved in regions of immobile ephrinA1 (*Figure 5C* and *Video 4*). The average maximal intensity of F-tractin-EGFP increased by about 80% in EphA2 clusters compared with immobile regions (*Figure 5D*, *Figure 5—source data 1*), revealing the extent to which EphA2 clustering enhances signaling to actin polymerization. Single-molecule imaging confirmed clustering of NCK-mEos3.2 and N-WASP-mEos3.2 in mobile ephrinA1 corrals (*Figure 5E*). The molecular dwell times also increased consistently in all measured cells in EphA2 clusters. On average, it increased NCK dwell time from 143 ms in immobile areas to 165 ms in mobile areas (*Figure 5F*), and N-WASP dwell time from 140 ms to 156 ms (*Figure 5G*). Therefore the live-cell measurement is in agreement with the in vitro reconstitution experiment (*Case et al., 2019*), that the increased dwell time of N-WASP is important for its activation.

## EphA2 clustering increases Grb2 and N-WASP dwell time in COS7 cells

The model cell line MDA-MB-231 expresses a very high level of EphA2 receptors, which may raise concern whether the observed dwell time difference is due to EphA2 pre-clustering. We validated these results by testing another cell line COS7, which expresses EphA2 at a low to moderate level (*Sabet et al., 2015*). EphrinA1 clustering and subsequent Grb2 and N-WASP recruitment upon cell contact are readily observed in mobile membrane corrals (*Figure 6A*). The clustering consistently increases Grb2-tdEos dwell time in all measured cells (averaged 241 ms in immobile areas to 310 ms in mobile areas) (*Figure 6B*), and increases N-WASP-mEOS3.2 dwell time in most cells only with a rare exception (averaged 214 ms in immobile areas to 235 ms in mobile areas) (*Figure 6C*).

## Discussion

Receptor clustering lies at the center of incorporating extracellular signals into amplified cellular responses (*Bray et al., 1998*; *Cebecauer et al., 2010*; *Taylor et al., 2017*). In this respect, lateral diffusion of membrane receptors is critical in controlling their clustering and thus the assembly of signaling complexes. The microfabrication technology described here provides a straightforward way to produce synthetic substrates displaying mobile and immobile ligands for monitoring clustering-dependent receptor signaling in individual cells. The utility of this technology was clearly

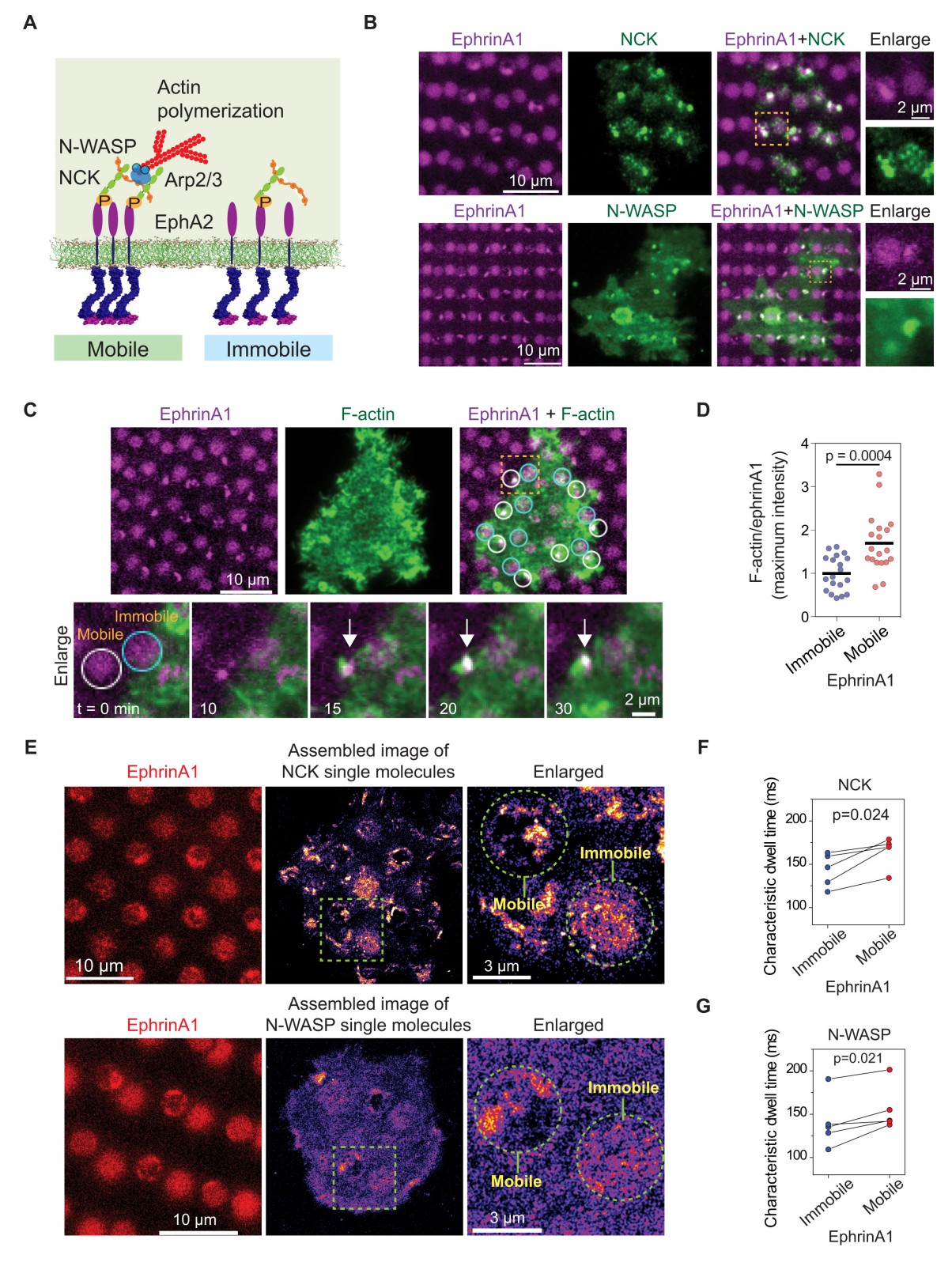

**Figure 5.** EphA2 clustering enhances NCK:N-WASP induced actin polymerization and increases molecular dwell time. (**A**) Schematic illustration of the signaling pathway from EphA2 receptor to actin polymerization. (**B**) Representative live-cell images of an NCK-mEOS3.2 or N-WASP-mEOS3.2 transfected cell spreading on the substrate after 1 hr. (**C**) Representative live-cell images of an F-tractin-EGFP transfected cell spreading on the substrate, with white circles indicating mobile ephrinA1 corrals and cyan circles indicating immobile ephrinA1. The marked yellow square is enlarged to

*Figure 5 continued on next page*

*Figure 5 continued*

highlight temporal actin polymerization dynamics with referring to mobile or immobile ephrinA1 contact. The white arrow indicates local actin polymerization. (D) Quantification of maximum F-tractin-EGFP intensities in mobile or immobile ephrinA1 corrals, normalized with ephrinA1 ligand intensity in each corral. Significance is analyzed by Student's t-test. N=19 cells. (E–G) Single-molecule imaging and quantification of NCK-mEos3.2 and N-WASP-mEos3.2. Condition same as *Figure 3F–I*. N=5 cells. Significance is analyzed by paired-group Student's t-test.

The online version of this article includes the following source data for figure 5:

**Source data 1.** Quantification of maximum F-tractin-EGFP/ephrinA1 intensity ratio in mobile or immobile ephrinA1 regions.

demonstrated by the observation of increased phosphorylation of clustered EphA2 compared to the non-clustered ones. Our results indicate that ephrinA1 ligand binding alone, without subsequent clustering, is not sufficient to induce or maintain EphA2 receptor phosphorylation. The increased local density of ephrinA1:EphA2 complexes, within clusters as occurs on mobile membrane corrals, is likely to promote trans-phosphorylation and activation of the receptor kinase (*Wiesner et al., 2006*). It is also possible that the clustered receptors could much more rapidly and efficiently re-phosphorylate each other after dephosphorylation. Additionally, phosphatases may be physically excluded from EphA2 clusters, in a manner similar to that observed in the immunological receptors (*Davis and van der Merwe, 2006*; *James and Vale, 2012*). Possibly a combination of these mechanisms gives rise to the amplified EphA2 activation when they are clustered. While these concepts have been previously considered (*Davis et al., 1994*; *Janes et al., 2012*), the methods we present here enable quantification of the effects of clustering on various aspects of the signaling pathway.

Two recent studies on LAT:Grb2:SOS and Nephrin:NCK:N-WASP systems proposed a kinetic proofreading model in biomolecular condensates (*Case et al., 2019*; *Huang et al., 2019*). The key principle is that there is a time lag between molecule recruitment and activation. Therefore, molecular timing functions as an important gating mechanism to drive non-equilibrium signaling responses: the long-dwelled molecules in phase-separated clusters are likely to be more productive in multi-step enzymatic reactions. This model was primarily inferred from simplified in vitro reconstitution experiments. Here, we found that both Grb2:SOS and NCK:N-WASP signaling molecules are condensed in EphA2 clusters in living cells, possibly forming a two-dimensional network. Importantly, these molecules dwelled longer inside the clusters, which is in agreement with the kinetic proofreading model. Only side-by-side comparison of mobile and immobile receptors made it possible to detect such differences in living cells.

However, the live cell results still differ from reconstitutions to some extent. The increase of molecular dwell times measured here in the receptor clusters in cells is smaller than observed in similar reconstitution studies (*Case et al., 2019*; *Huang et al., 2016*). There are several possible reasons for this. (1) EphA2 clusters may not be directly comparable with LAT/Nephrin clusters. EphA2 also has multiple phosphotyrosine sites (*Fang et al., 2008*), but the physical organization of molecules inside the clusters remains to be elucidated. (2) It is possible that other factors, such as phosphatases, present in the live cell modulate the overall kinetic behavior. (3) The most likely explanation, perhaps, is that the average dwell time simply fails to capture the relevant aspect of the system. In the case of Ras activation by SOS, signaling activity is predominantly driven by a small number of activated molecules and is not reflective of the average (*Huang et al., 2019*; *Huang et al., 2016*). Small numbers of very long dwelling molecules would not likely be observed in these live-cell experiments due to photobleaching and other imaging noise sources. The absolute dwell

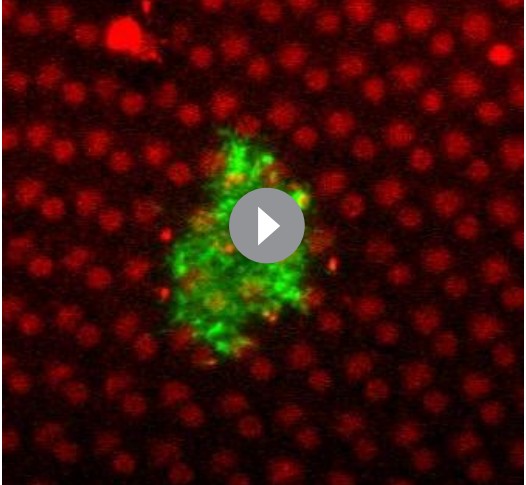

**Video 4.** An F-tractin-EGFP transfected MDA-MB-231 cell spreading on ephrinA1 micropatterned substrate. https://elifesciences.org/articles/67379#video4

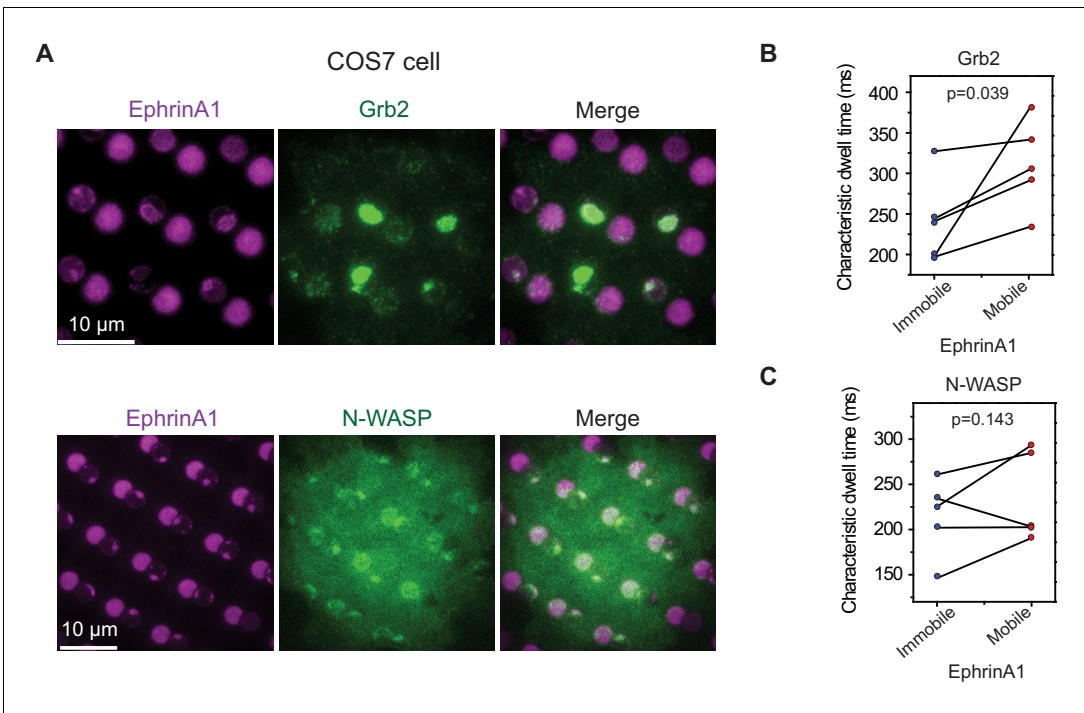

**Figure 6.** EphA2 clustering increases Grb2 and N-WASP dwell time in COS7 cells. (**A**) Representative live-cell images of Grb2-tdEOS or N-WASP-mEOS3.2 transfected COS7 cells spreading on the hybrid substrate. Quantification of (**B**) Grb2-tdEos or (**C**) N-WASP-mEos3.2 dwell time in COS7 cells. Condition same as *Figure 3F–I*. N=5 cells. Significance is analyzed by paired-group Student's t-test.

time is almost impossible to be deconvolved from photobleaching effects in current live-cell measurements, given the fact that the optical performance of fluorescent proteins cannot match organic dyes used in vitro (*Chen et al., 2021*). Higher spatial and temporal resolution is still needed to delineate more detailed mechanisms of how biomolecular condensation modulates signaling efficiency.

In conclusion, EphA2 clustering enhances both receptor phosphorylation and binding dwell time of downstream signaling molecules in living cells, providing further evidence to support the kinetic proofreading mechanism of molecular activation in biomolecular condensates. While we have investigated EphA2 receptors in the current study, the technology described here could be extended to investigate other membrane-localized receptor systems. Further, the substrates developed here could be directly utilized in the screening of pharmacological agents to target diseases that involve clustering-dependent receptor signaling (*Chang et al., 2018*; *Lohmüller et al., 2013*; *Salaita et al., 2010*).

# Materials and methods

## Key resources table

| Reagent type (species) or resource | Designation | Source or reference | Identifiers | Additional information |
|---|---|---|---|---|
| Cell line (*Homo-sapiens*) | MDA-MB-231 | ATCC | ATCC Cat# CRL-12532, RRID:CVCL_0062 | |
| Cell line (*Homo-sapiens*) | PC-3 | ATCC | ATCC Cat# CRL-7934, RRID:CVCL_0035 | |

*Continued on next page*

*Continued*

| Reagent type (species) or resource | Designation | Source or reference | Identifiers | Additional information |
|---|---|---|---|---|
| Cell line (*Chlorocebus aethiops*) | COS-7 | ATCC | ATCC Cat# CRL-1651, RRID:CVCL_0224 | Provided by UCB Cell Culture Facility |
| Antibody | Anti-EphA2 (Rabbit monoclonal) | Cell Signaling Technology | Cat# CST 6997 RRID:AB_10827743 | IF(1:1000) |
| Antibody | Anti-phospho-p44/42 MAPK (Erk1/2) (Rabbit polyclonal) | Cell Signaling Technology | Cat# CST 9101 RRID:AB_331646 | IF(1:1000) |
| Antibody | Anti-phospho-EphA2 (Tyr588) (D7X2L) (Rabbit monoclonal) | Cell Signaling Technology | Cat# CST 12677 RRID:AB_2797989 | IF(1:1000) |
| Recombinant DNA reagent | Grb2-tdEos (plasmid) | DOI: 10.1073/pnas.1203397109 | | |
| Recombinant DNA reagent | SOS-tdEos (plasmid) | DOI: 10.1073/pnas.1203397109 | | |
| Recombinant DNA reagent | CAAX-tdEos (plasmid) | DOI: 10.1073/pnas.1203397109 | | |
| Recombinant DNA reagent | F-tractin-EGFP (plasmid) | This paper | | Maintained in Ronen Zaidel-Bar lab |
| Recombinant DNA reagent | NCK-mEos3.2 (plasmid) | This paper | | Cloned in Mechanobiology Institute Core facility, National University of Singapore |
| Recombinant DNA reagent | N-WASP-mEos3.2 (plasmid) | This paper | | Cloned in Mechanobiology Institute Core facility, National University of Singapore |

## Patterned substrate of mobile and immobile ligands

Phospholipid vesicles were prepared using previously published methods (*Galush et al., 2008*). 96% of DOPC (1,2-dioleoyl-sn-glycero-3-phosphocholine) was mixed with 4% of Ni-NTA-DOGS (1,2-dioleoylsn-glycero-3-[(N-(5-amino-1-carboxypentyl)iminodiacetic acid)-succinyl]) (Avanti lipids) to form supported membranes. The pattern is fabricated by UV lithography. Briefly, clean coverslips were incubated with PLL-(*g*)-PEG-biotin (50%, Susos) for at least 2 hr. The polymer-coated coverslips were then etched by deep UV (UVO-cleaner 342-220, Jelight) with a designed photomask to generate patterns. Lipid vesicles were then deposited on the coverslips to form supported membranes on the etched surface only. Later, the substrates were blocked by 0.05% bovine serum albumin (BSA; Sigma-Aldrich) for 2 hr, and then incubated with 1 µg/ml of DyLight-405 NeutrAvidin (Thermo Fisher Scientific) for 30 min to bind biotin. After rinse, the substrates were then incubated with a solution of 5 nM of ephrinA1-Alexa 680 and 1 µg/ml of RGD-PEG-PEG-biotin (Peptides International) for 60 min for surface functionalization. EphrinA1 was expressed with a C-terminal 10-histidine tag and purified from insect cell culture (*Xu et al., 2011*), and labeled with Alexa 680 fluorophore (Thermo Fisher Scientific) followed by vendor's manual. Finally, the protein incubation solutions were exchanged with imaging buffer (25 mM Tris, 140 mM NaCl, 3 mM KCl, 2 mM $CaCl_2$, 1 mM $MgCl_2$, and 5.5 mM D-glucose) and warmed up to 37°C prior addition of cells.

The three-component substrates were prepared by two steps of UV etching. After the first etch on PLL-(*g*)-PEG-biotin surface, the coverslip was then coated with PLL-(*g*)-PEG-NTA or PLL-*g*-PEG (Susos), and underwent another etch to form supported membranes. The two etch processes were overlaid randomly for regularly circular patterns, or aligned under the microscope to obtain high accuracy. The substrate was then blocked by BSA and functionalized with ligands similar as above. To modulate immobile/mobile ephrinA1 intensity ratio, PLL-(*g*)-PEG-NTA was titrated by mixing

with PLL-(*g*)-PEG. The fluorescence intensity of ephrinA1 is measured 30 mins after washing out proteins in solution, at a similar time when the cellular experiments are performed.

## Cell culture, plasmid, and immunostaining

MDAMB-231 cells (ATCC) and PC-3 cells (ATCC) were grown in DMEM (high glucose) (Thermo Fisher Scientific) medium. COS-7 cells (UC Berkeley Cell Culture Core Facility) were grown in RPMI 1640 (Thermo Fisher Scientific) medium. The medium is supplemented with 10% fetal bovine serum (FBS) (Thermo Fisher Scientific) and 1% penicillin/streptomycin (Thermo Fisher Scientific). They are authenticated using STR profiling by ATCC, and they are tested to be free from mycoplasma contamination. Cells were detached by enzyme-free dissociation buffer (Thermo Fisher Scientific) and then allowed to interact with indicated substrates. Cells were transfected by Lipofectamine 2000 (Thermo Fisher Scientific). Grb2-tdEos, SOS-tdEos, and CAAX-tdEos plasmids are the same as previously published (*Oh et al., 2012*). F-tractin-EGFP is maintained in Zaidel-Bar's lab. NCK-mEos3.2 and N-WASP-mEos3.2 are cloned in Mechanobiology Institute Core Facility. For immunostaining, cells were fixed with 4% paraformaldehyde, and permeabilized with 0.1% Triton-X, followed by standard immunostaining protocol. Primary antibodies include rabbit anti-EphA2 (CST, 6997), rabbit anti-pErk (CST, 9101), and rabbit anti-pY588-EphA2 (CST, 12677). The secondary antibody is goat anti-rabbit antibody conjugated with Alexa 488 fluorophores (Thermo Fisher Scientific).

## Live-cell and single-molecule imaging

An Eclipse Ti Inverted Microscope (Nikon) with a TIRF system and Evolve EMCCD Camera (Photometrics) were used for live-cell imaging. TIRF microscopy was performed with a 100× TIRF objective with a numerical aperture of 1.49 (Nikon) and an iChrome MLE-L multilaser engine as a laser source (Toptica Photonics). Immunofluorescent imaging was also acquired in an Eclipse Ti Inverted Microscope (Nikon) with CSU-X1 confocal spinning disk unit (Yokogawa).

Time-lapse single-molecule imaging of Grb2-tdEos, SOS-tdEos, NCK-mEos3.2, and N-WASP-mEos3.2 were performed by TIRF microscopy, in a way such as to optimize signal-to-noise and temporal resolution by coupling minimizing laser power and maximizing video rate. To increase tracking accuracy, the density of individual molecules was controlled by 405 nm laser illumination to be about ~0.5/μm². Far-red channel (ex=647 nm, em>655 nm) were acquired before single-molecule recording to localize mobile and immobile ephrinA1 corrals. The autofluorescence on the red channel was completely photobleached before photo-switching Eos by a 405 nm beam. After photo-switching, a small amount of Eos molecules were visualized and recorded by EMCCD with 20 frames per s video rate. Each movie contains 1000 frames for further analysis. Membrane localized CAAX-tdEos movies were used to calculate photobleach rate, acquired at the same microscopic setup.

## Image analysis

Live-cell and immunofluorescence images were analyzed to quantify Grb2-tdEos, F-tractin-EGFP, anti-EphA2, and anti-pY588-EphA2 intensities in mobile and immobile ephrinA1 regions. The regions of mobile and immobile ephrinA1 were outlined to generate masks, so the average intensity of different channels can be measured in the same corral and the ratios were calculated after subtraction of noises.

For live-cell single-particle tracking, a cross-correlation single particle tracking method was used to determine the centroid positions of tdEos or mEos3.2 single molecules (*Oh et al., 2012*; *Oh et al., 2014*). A trajectory was created by connecting the subsequent xy coordinates through the frames using the nearest neighbor method. The first positions of each trajectory were assembled to generate a localization image. By using pre-acquired ephrinA1 image as a mask, single-molecule signals coming from mobile or immobile ephrinA1 regions were separated to calculate spatially resolved binding kinetics. The dwell time distributions of molecules in the two different regions were fitted with a two-order exponential decay function $y=y_0+A_1e^{-t/\tau_1}+A_2e^{-t/\tau_2}$, providing two characteristic time constants $\tau_1$ and $\tau_2$. We noticed the data points become noisy for longer dwell time but this population represents less than 0.5% of the total molecules. We applied this model to fit the vast majority of the data. To be consistent $\tau_2$ was used for pairwise comparison of each cell. It is inevitable to avoid short-lived fluorescence events in single-molecule imaging due to the intrinsic power law of fluorescence on/off time. Furthermore, the non-bound cytoplasmic molecules may

enter into the evanescent field and go out very quickly. These factors contribute to the short decay time. The second order of exponential decay function is not perfect to estimate the off-rate but it represents the statistic value in our system.

## Acknowledgements

The authors thank Professor Michael Sheetz for stimulating discussions. This work was supported by the National Institutes of Health, National Cancer Institute Physical Sciences in Oncology Network Project 1-U01CA202241. Collaborative work at the Mechanobiology Institute, National University of Singapore, was supported by CRP001-084. ZC is also funded by Shanghai Municipal Science and Technology Major Project ZJLab (2018SHZDZX01).

## Additional information

### Funding

| Funder | Grant reference number | Author |
| --- | --- | --- |
| National Cancer Institute | | Zhongwen Chen<br>Jay T Groves |
| National Research Foundation Singapore | CRP001-084 | Zhongwen Chen<br>Dongmyung Oh<br>Ronen Zaidel-Bar<br>Jay T Groves |
| Shanghai Municipal Science and Technology Commission | ZJLab (2018SHZDZX01) | Zhongwen Chen |
| Novo Nordisk Foundation | Challenge Grant Center for Geometrically Engineered Cellular Systems | Jay T Groves |

The funders had no role in study design, data collection and interpretation, or the decision to submit the work for publication.

### Author contributions

Zhongwen Chen, Conceptualization, Data curation, Formal analysis, Investigation, Visualization, Methodology, Writing - original draft; Dongmyung Oh, Formal analysis, Investigation, Methodology; Kabir Hassan Biswas, Formal analysis, Writing - review and editing; Ronen Zaidel-Bar, Supervision, Funding acquisition, Writing - review and editing; Jay T Groves, Conceptualization, Supervision, Funding acquisition, Writing - review and editing

### Author ORCIDs

Zhongwen Chen https://orcid.org/0000-0001-5218-0152
Dongmyung Oh https://orcid.org/0000-0003-0817-5254
Kabir Hassan Biswas http://orcid.org/0000-0001-9194-4127
Jay T Groves https://orcid.org/0000-0002-3037-5220

### Decision letter and Author response

Decision letter https://doi.org/10.7554/eLife.67379.sa1
Author response https://doi.org/10.7554/eLife.67379.sa2

## Additional files

### Supplementary files
• Transparent reporting form

## Data availability

All data generated or analysed during this study are included in the manuscript and supporting files. Source data files have been provided for Figures 2, 3, 4, and 5.

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
