## [Decision Letter]

**Acceptance summary:**

This is a technically innovative study and will be of broad interest to the cell signaling and also biomolecular condensate communities.

**Decision letter after peer review:**

Thank you for submitting your article "Probe the effect of clustering on EphA2 receptor signaling efficiency by subcellular control of ligand-receptor mobility" for consideration by *eLife*. Your article has been reviewed by 3 peer reviewers, one of whom is a member of our Board of Reviewing Editors, and the evaluation has been overseen by Jonathan Cooper as the Senior Editor.

Essential revisions:

1) More detailed quantification of patch behavior and molecules within patches: Given the variability between patches of clusters in behavior between cells, please provide more detailed quantification of patches within a single cells. Ideally, this would include further analyses including a clustering estimate (as suggested by reviewer 1), tracking particles to estimate diffusivity (as outlined by reviewer 2).

2) Magnitude and mechanism of effects via phase separation: The reviewers all felt that the effects of clustering on dwell times were more modest than might have been expected based on in vitro systems and also that it isn't definite that the impacts are due to a phase change/physiochemical effects. We suggest bolstering the conclusion that changes in physiochemical properties result in changes to dwell time and signal transduction, by starting with immobile substrates that are approximately equal to the density of the clusters formed with mobile substrates. If a difference remains, then there would be more weight to the authors' claim that a change in physiochemical properties contributes to changed behavior of signaling molecules. Please also more directly discuss the implications of the discrepancy of the impact of clustering on dwell times compared to previous in vitro experiments.

3) Downstream-signaling impacts: Ideally, we would like to see some read-out of of the downstream signaling pathways such as Erk nuclear localization. This may involve new experiments where single cells are on mobile vs immobile substrates at similar densities as in the patterned substrates reported here. We think these experiments could substantially strengthen the revision but will consider a revision without these if they are too involved. If however you have data that addresses the downstream activity of signaling, we strongly encourage you to incorporate it.

4) Please address all figure-related and textual issues outlined and suggested by all reviewers.

*Reviewer #1 (Recommendations for the authors):*

Point 1: In Figure 1, clustering of ephrinA1 by the cell in mobile corrals is certainly visually apparent. However, some quantification of clustering within these regions would also be beneficial. Can the authors perform some simple image analysis to provide a clustering metric, like the one in panel 3C, for example, and use it to compare ephrinA1 clustering in mobile and immobile regions?

Point 2: Similar to point 1, EphA2 appears clearly clustered in mobile regions in panel 2A. Please provide some clustering quantification of EphA2 in mobile vs. immobile regions. Also, it appears that ephrinA1 and EphA2 are nearly perfectly co-localized in the upper images, while ephrinA1 and pY588-EphA2 are not necessarily as co-localized. Specifically, it appears that pY588-EphA2 occasionally shows some clusters that do not overlap with the ephrinA1 clusters. Why is this? Is this an artifact of the fixing/immunostaining procedure, or can pY588-EphA2 cluster via other mechanisms? Please also provide merged images of the ephrinA1 and EphA2 or pY588-EphA2 channels to show co-localization. Since the merged images will be shown, please adjust the colors to magenta/green for colorblind readers.

Point 3: In panel 3F, the assembled SOS image seems to show a ring of SOS around the edge of the mobile corrals, while Grb2 forms clusters that seem to more clearly co-localize with ephrinA1 clusters. Why does Grb2 form clusters while SOS forms a different structure? Also, the ephrinA1 mobile corrals corresponding to the SOS image do not appear strongly clustered in 3F, though these corrals do appear clustered in the Grb2 image set. Why is ephrinA1 not clustered with SOS in these images?

*Reviewer #2 (Recommendations for the authors):*

1. The description on p. 9 of the effect of clustering on Grb2 recruitment seems somewhat overstated-if the increase in receptor autophosphorylation is 60% in clustered vs. unclustered patches, the difference in Grb2 binding of 80% is not very "remarkable," and whether this difference is meaningful or significant deserves some discussion. In other words, could all of the binding differences be explained by differences in receptor autophosphorylation?

2. in Figure 3F (spt-PALM imaging of Grb2 recruitment), it is interesting to me that the sites of binding appear in many cases to be ring-shaped (that is, concentrated on the outer periphery of the patch), while the clustered receptor itself does not show the same pattern. Some discussion of this and possible mechanisms to explain it would be helpful.

3. The authors should also mention why two decay constants are used to fit dwell time distributions (p. 16) and why only one of these is used in Figure 3H.

*Reviewer #3 (Recommendations for the authors):*

1) As constructed, the experiments start with an equal density of ephrinA1 on mobile and immobile substrates. However, upon clustering, the density of ephrinA1:Eph2A in mobile clusters becomes higher in the clustered region than in the immobile region. This clearly illustrates the importance of the rearrangement of Eph receptors on the cell membrane to drive cellular signal transduction. In lines 184-185, the authors claim that there is a physiochemical change (phase separation?) brought about by clustering that contributes to the different results observed between mobile and immobile receptors. However, a direct link between physiochemical properties and signaling is not evident from the experiments performed. The reported differences reported in the ensuing experiments could simply be due to changes in density of receptors attached to mobile substrates. To confirm that there are indeed changes in physiochemical properties that result in changes to dwell time and signal transduction, the authors should start with immobile substrates that are approximately equal to the density of the clusters formed with mobile substrates. If a difference remains, then there would be more weight to the authors' claim that a change in physiochemical properties contributes to changed behavior of signaling molecules. This is important because of the differences in valency between previously investigated clusters, as both LAT and nephrin are multivalent with 3 binding sites for adaptor proteins, and Eph2A which only has a single site. Eph2A clusters may simply be oligomers that share physiochemical properties with the surrounding environment with downstream changes arising from density changes while LAT and nephrin are bona fide phase separated condensates with unique physiochemical properties.

2) The comparison in Figure 5 is a bit strange because the observed molecules are not analogous. Grb2 and Nck both bind to Eph2A while SOS and N-WASP bind to the adaptor proteins. It would be better to compare the behaviors of either Grb2 and Nck or SOS and N-WASP.

---

## [Author Response]

Essential revisions:1) More detailed quantification of patch behavior and molecules within patches: Given the variability between patches of clusters in behavior between cells, please provide more detailed quantification of patches within a single cells. Ideally, this would include further analyses including a clustering estimate (as suggested by reviewer 1), tracking particles to estimate diffusivity (as outlined by reviewer 2).

Based on the reviewer’s suggestions, we have included more detailed quantification of patch behavior.

1) We use ‘standard deviation’ of ephrinA1 intensity, or EphA2/pY588EphA2 intensity in each bilayer corral as a measureable parameter to show cluster variability.

2) We quantified the apparent Grb2 diffusion rate in the mobile and immobile areas.

2) Magnitude and mechanism of effects via phase separation: The reviewers all felt that the effects of clustering on dwell times were more modest than might have been expected based on in vitro systems and also that it isn't definite that the impacts are due to a phase change/physiochemical effects. We suggest bolstering the conclusion that changes in physiochemical properties result in changes to dwell time and signal transduction, by starting with immobile substrates that are approximately equal to the density of the clusters formed with mobile substrates. If a difference remains, then there would be more weight to the authors' claim that a change in physiochemical properties contributes to changed behavior of signaling molecules. Please also more directly discuss the implications of the discrepancy of the impact of clustering on dwell times compared to previous in vitro experiments.3) Downstream-signaling impacts: Ideally, we would like to see some read-out of of the downstream signaling pathways such as Erk nuclear localization. This may involve new experiments where single cells are on mobile vs immobile substrates at similar densities as in the patterned substrates reported here. We think these experiments could substantially strengthen the revision but will consider a revision without these if they are too involved. If however you have data that addresses the downstream activity of signaling, we strongly encourage you to incorporate it.

We thank for the reviewer’s valuable comments and suggested experiments, a version of which we have done and now included in the revised paper. Specifically, we have now included nuclear Erk activation data comparing mobile and immobile ephrinA1 substrates. Immunostaining of the PC-3 cells with a phospho-specific anti-Erk antibody revealed ~50% increase in the levels of activated nuclear Erk in cells seeded on mobile ephrinA1 substrates compared to the immobile ones.

The reviewers also raise a much deeper question, of broad significance across the entire field of signal transduction, concerning how much of a physical modulation (e.g. changes in dwell time distribution in this case) is necessary to alter a signal. The newly added experiments mentioned above at least establish that the physical changes induced by mobility restriction propagate downstream to Erk activation. However, it was never our intention to assert that changes in dwell time distribution were uniquely responsible for this. Rather, we are relating the cellular experimental observations here to recent reconstitution experiments that have shown how rare, long dwelling SOS molecules may be primarily responsible for transmitting signals to Ras (*Science* 2019, 363: 1098; *PNAS* 2016, 29: 8218). The point is that the cellular results are consistent with the reconstitution results, but we believe far more work—spanning many aspects of the field of signal transduction and protein condensates—is required to establish proven causal relationships. We have clarified text in the discussion on our interpretation of this data.

4) Please address all figure-related and textual issues outlined and suggested by all reviewers.

We have addressed all figure-related and textual issues carefully. Note we also reorganized the figures to include new data and quantifications.

Reviewer #1 (Recommendations for the authors):Point 1: In Figure 1, clustering of ephrinA1 by the cell in mobile corrals is certainly visually apparent. However, some quantification of clustering within these regions would also be beneficial. Can the authors perform some simple image analysis to provide a clustering metric, like the one in panel 3C, for example, and use it to compare ephrinA1 clustering in mobile and immobile regions?

We agree that ephrinA1 clusters have a great degree of variability. When initially contact by the cell, almost all ephrinA1 molecules inside each membrane patch will come together to form a big cluster. Later, the big cluster is also dynamic, moving from one side to another, or divide into several smaller clusters and moving around. Because of the spatial confinement, those clusters tend to distribute in the periphery of the membrane corral. To better present the dynamic nature of ephrinA1 clustering in our system, we have included snapshots of an ephrinA1 membrane corral before and after cell contact for 30mins (Figure 1C). We also included a quantification of ‘average’ and ‘standard deviation’ of ephrinA1 intensity of each membrane corral underneath the cell (Figure 1 —figure supplement 1D). The standard deviation indicates pixel-to-pixel variation of ephrinA1 intensity. Because of the variation of ephrinA1 clusters, the standard deviation shows a large fluctuation among different corrals compared with uniform distribution in un-contact (un-clustered) corrals, although they all have the same average intensity.

In immobile regions, the ephrinA1 intensity and distribution won’t change after cell contact.

Text revision: ‘The cluster is also dynamic, moving from one side to another, or divide into several smaller clusters and moving around, resulting in a great degree of variability as shown by the large fluctuation in the standard deviations of ephrinA1 intensity in each membrane corral (Figure 1 —figure supplement 1D). Because of the spatial confinement, those clusters tend to be trapped in the periphery of the corrals.’

Point 2: Similar to point 1, EphA2 appears clearly clustered in mobile regions in panel 2A. Please provide some clustering quantification of EphA2 in mobile vs. immobile regions. Also, it appears that ephrinA1 and EphA2 are nearly perfectly co-localized in the upper images, while ephrinA1 and pY588-EphA2 are not necessarily as co-localized. Specifically, it appears that pY588-EphA2 occasionally shows some clusters that do not overlap with the ephrinA1 clusters. Why is this? Is this an artifact of the fixing/immunostaining procedure, or can pY588-EphA2 cluster via other mechanisms? Please also provide merged images of the ephrinA1 and EphA2 or pY588-EphA2 channels to show co-localization. Since the merged images will be shown, please adjust the colors to magenta/green for colorblind readers.

We use ‘standard deviation’ as a measurable parameter for clustering quantification, to compare clustered vs non-clustered EphA2 and pY588-EphA2 in Figure 1 —figure supplement 2. The results show that EphA2 and pY588-EphA2 intensities have larger fluctuations in mobile region, indicating their clustering behavior and cluster variability.

Text revision: ‘Consistent with ephrinA1 cluster variability, EphA2 and pY588-EphA2 intensities also showed larger fluctuation in mobile regions compared with immobile ones (Figure 1 —figure supplement 2).’

We have also included merged images of ephrinA1 and EphA2/pY588-EphA2 with magenta/green in Figure 2. The ephrinA1 clusters tend to locate in the periphery of the mobile patch because of the spatial confinement. EphrinA1 and pY588-EphA2 still largely co-localize in mobile regions, especially around the bilayer ring. One thing to notice is that ephrinA1 intensity correlates very well with EphA2 in immunostaining, but it does not necessarily linearly correlate with pY588-EphA2 in the images, which may suggest the phosphorylation and dephosphorylation dynamic of EphA2 receptors. We are not aware of any knowledge of pY588-EphA2 clustering by other mechanisms.

Point 3: In panel 3F, the assembled SOS image seems to show a ring of SOS around the edge of the mobile corrals, while Grb2 forms clusters that seem to more clearly co-localize with ephrinA1 clusters. Why does Grb2 form clusters while SOS forms a different structure? Also, the ephrinA1 mobile corrals corresponding to the SOS image do not appear strongly clustered in 3F, though these corrals do appear clustered in the Grb2 image set. Why is ephrinA1 not clustered with SOS in these images?

As stated above, sometimes the clusters are likely to locate around the edge of mobile corrals. This has been observed for both Grb2 and SOS.

The previous image of ephrinA1 is possibly out of focus so they do not appear strongly clustered. To avoid confusion, we have replaced the image with a clearer one in the new Figure3F. EphrinA1 clustering in mobile corrals is very robust—nearly 100%

Reviewer #2 (Recommendations for the authors):1. The description on p. 9 of the effect of clustering on Grb2 recruitment seems somewhat overstated-if the increase in receptor autophosphorylation is 60% in clustered vs. unclustered patches, the difference in Grb2 binding of 80% is not very "remarkable," and whether this difference is meaningful or significant deserves some discussion. In other words, could all of the binding differences be explained by differences in receptor autophosphorylation?

We tuned the tone to ‘a significant difference was observed in the local recruitment of Grb2 to mobile or immobile ephrinA1:EphA2 complexes as cells spread’. To address the difference between receptor phosphorylation (~60%) and Grb2 recruitment (~80%), we have included Grb2 single molecule on-rate measurement in Figure S6D. The results show that clustering increases Grb2 association rate (Kon) by about 60% compared with immobile ephrinA1, a value commensurate with the increase in the level of phosphorylated EphA2. As a result, the increase of 80% Grb2 total recruitment is a combination of increased receptor phosphorylation and elongated dwell time.

Text revision: ‘By counting molecule numbers in mobile or immobile regions, we found that clustering increases Grb2 association rate (Kon) by about 60%, a value commensurate with the increase of phosphorylated EphA2 (Figure 3 —figure supplement 1D). As a result, the increase of 80% Grb2 total recruitment is a combination of increased receptor phosphorylation and elongated dwell time.‘

2. In Figure 3F (spt-PALM imaging of Grb2 recruitment), it is interesting to me that the sites of binding appear in many cases to be ring-shaped (that is, concentrated on the outer periphery of the patch), while the clustered receptor itself does not show the same pattern. Some discussion of this and possible mechanisms to explain it would be helpful.

This is related to reviewer 1 point 1. The clusters tend to locate in the periphery of the bilayer patch because that the ligand: receptor complex move on the membrane and they are blocked by the pattern boundary. We believe Grb2 still co-localize with the receptors and ephrinA1 ligands.

3. The authors should also mention why two decay constants are used to fit dwell time distributions (p. 16) and why only one of these is used in Figure 3H.

We have now added the explanation in the Methods.

Text revision: ‘It is inevitable to avoid short-lived fluorescence events in single molecule imaging due to the intrinsic power law of fluorescence on/off time. Furthermore, the non-bound cytoplasmic molecules may enter into the evanescent field and go out very quickly. These factors contribute to the short decay time. The second order of exponential decay function is not perfect to estimate the off rate but it represents the statistic value in our system.‘

Reviewer #3 (Recommendations for the authors):1) As constructed, the experiments start with an equal density of ephrinA1 on mobile and immobile substrates. However, upon clustering, the density of ephrinA1:Eph2A in mobile clusters becomes higher in the clustered region than in the immobile region. This clearly illustrates the importance of the rearrangement of Eph receptors on the cell membrane to drive cellular signal transduction. In lines 184-185, the authors claim that there is a physiochemical change (phase separation?) brought about by clustering that contributes to the different results observed between mobile and immobile receptors. However, a direct link between physiochemical properties and signaling is not evident from the experiments performed. The reported differences reported in the ensuing experiments could simply be due to changes in density of receptors attached to mobile substrates. To confirm that there are indeed changes in physiochemical properties that result in changes to dwell time and signal transduction, the authors should start with immobile substrates that are approximately equal to the density of the clusters formed with mobile substrates. If a difference remains, then there would be more weight to the authors' claim that a change in physiochemical properties contributes to changed behavior of signaling molecules. This is important because of the differences in valency between previously investigated clusters, as both LAT and nephrin are multivalent with 3 binding sites for adaptor proteins, and Eph2A which only has a single site. Eph2A clusters may simply be oligomers that share physiochemical properties with the surrounding environment with downstream changes arising from density changes while LAT and nephrin are bona fide phase separated condensates with unique physiochemical properties.

We would like to clarify that we don’t directly refer the physicochemical changes in the EphA2 clusters to phase separations, in the same way to LAT: Grb2: SOS or Nephrin: NCK: N-WASP systems. Here, the physicochemical changes mean that mobility-allowed receptor clustering enhances EphA2 autophosphorylation, and in the meanwhile condensates its downstream molecules. The molecular dwell time is an immediate parameter that can be reliably measured and compared between clustered and un-clustered receptors. However, it was never our intention to assert that changes in dwell time distribution were uniquely responsible for the downstream effects. Rather, we are relating the cellular experimental observations here to recent reconstitution experiments that have shown how rare, long dwelling SOS molecules may be primarily responsible for transmitting signals to Ras (*Science* 2019, 363: 1098; *PNAS* 2016, 29: 8218). The point is that the cellular results are consistent with the reconstitution results.

The increase of molecular dwell time could possibly come from multivalent binding (EphA2 has 4 phosphor-tyrosine sites: 2 in the juxtamembrane region, 1 in the kinase domain, and 1 in the tail. *J Biol Chem* 2008 283: 16017), or rebinding mechanism (density effect), or both. The detailed molecular physical organization and the source of dwell time variability remain to be elucidated.

Text revision: ‘These results suggest that clustering of ephrinA1:EphA2 complexes enabled by the mobile ligands resulted in a change in their physicochemical properties: the clustering enhances EphA2 autophosphorylation, and in the meanwhile condensates its downstream molecules.’

2) The comparison in Figure 5 is a bit strange because the observed molecules are not analogous. Grb2 and Nck both bind to Eph2A while SOS and N-WASP bind to the adaptor proteins. It would be better to compare the behaviors of either Grb2 and Nck or SOS and N-WASP.

We thank for the reviewer’s comment. It is true Grb2 and N-WASP are not analogous, but still they represent the key molecules in the cluster complexes to regulate MAPK signaling or actin polymerization pathway. We believe it does not affect the main conclusion of the paper.